# A CMM-Based Method of Control Point Position Calibration for Light Pen Coordinate Measuring System

**DOI:** 10.3390/s20195592

**Published:** 2020-09-29

**Authors:** Sen Wang, Shugui Liu, Qing Mao

**Affiliations:** State Key Laboratory of Precision Measuring Technology and Instruments, Tianjin University, Tianjin 300072, China; senwang@tju.edu.cn (S.W.); mao_qing@tju.edu.cn (Q.M.)

**Keywords:** coordinate measuring, vision metrology, light pen, control point position calibration, CMM, PnP problem, structure from motion

## Abstract

A light pen coordinate measuring system (LPCMS) is a kind of vision-based portable coordinate measuring technique. It implements coordinate measurement by analyzing the image of a light pen, which has several control points and a probe. The relative positions of control points need to be determined before measurement and serve as the measuring basis in LPCMS. How to accurately calibrate the relative positions of control points is the most important issue in system calibration. In this paper, a new method of control point position calibration based on a traditional coordinate measuring machine (CMM) is proposed. A light pen is fastened to the measuring arm of a CMM and performs accurate translational movement driven by the CMM. A camera is used to capture the images of control points at different positions, and the corresponding readings of the CMM are recorded at the same time. By establishing a separate coordinate system for each control point, the relative positions of the control points can be transformed to the differences of a series of translation vectors. Experiments show that the calibration repeatability of control point positions can reach 10 μm and the standard deviation of measurement of the whole LPCMS can reach 30 μm. A CMM is used to generate accurate translation, which provides a high accuracy basis of calibration. Through certain mathematical treatment, tremendous data acquired by moving the light pen to tens of thousands of different positions can be processed in a simple way, which can reduce the influence of random error. Therefore, the proposed method provides a high-accuracy solution of control point position calibration for LPCMS.

## 1. Introduction

Coordinate measuring is a key technique in manufacturing and reverse engineering [1]. With the development of the automobile industry, shipbuilding industry, and aerospace industry, coordinate measuring techniques are increasingly applied to the in situ measurement of large workpieces. Although a traditional coordinate measuring machine (CMM) has very high accuracy, it has poor flexibility and requires strict environmental conditions, which makes it not competent for in situ measurement [2,3]. Therefore, several kinds of portable coordinate measuring techniques have arisen in the last two decades, including laser scanners [4], laser trackers [5], photogrammetry-based devices [6], and so on. They have excellent flexibility, large measurement range, and relatively high accuracy. Among them, the light pen coordinate measuring system (LPCMS) is indispensible for its ability to measure the hidden features of a workpiece, including, but not limited, to deep holes [7,8,9].

LPCMS is typically composed of a camera, a light pen, a laptop, and some data transmission wires. The key component is the light pen, which has several control points and a probe on it. The control points are generally active luminous infrared light emitting diodes (LEDs). Once they are installed on the light pen, their relative positions are fixed. The fundamental principle of LPCMS is to determine the relative position and orientation between the light pen and the camera, according to the control point positions and the image of the control points. Therefore, it can be said that the relative positions of control points serve as the measuring basis in LPCMS, and the accuracy of the whole system is highly dependent on the calibration accuracy of the control point positions. The main issue of this paper is how to fulfill high accuracy calibration of the control point positions.

There have been some methods to calibrate control point positions, which can be classified into two categories: One kind is to measure the relative positions of the control points directly using an image measuring apparatus together with a traditional CMM [10]. However, the control point is not an ideal point but a luminous point of a certain size. Actually, what the calibration really needs to determine is the position of the luminous center of the LED, which cannot be accurately and directly located. Therefore, direct measurement methods can only obtain a rough result. The other kind is the vision-based structure-from-motion (SFM) method, which uses a moving camera to obtain a group of images of control points from different viewpoints [11]. Control point positions can be recovered up to a scale from this group of images using bundle adjustment. Then the absolute dimension is determined by a calibration bar that has accurate length. This is a nonlinear optimization process and does not need to explicitly point out the luminous center of the control points. The problem with this kind of method is that the calibration accuracy is strongly affected by the accuracy of the camera calibration. In order to obtain higher calibration accuracy, the camera needs to move to more viewpoints to give more information about control point positions. Each time a new camera viewpoint is added, the number of variables to be optimized increases by six. These variables represent just the position and orientation of the camera at different viewpoints, and have nothing to do with the relative positions of the control points. Therefore, this will make the nonlinear least squares problem rather complicated and the solving process unstable. In addition to this, the absolute dimension of control point positions are determined by only a few lengths on the calibration bar, which makes the result unreliable if the reference lengths are not accurate enough.

In this paper, a new high-accuracy method of control point position calibration is proposed. It utilizes a traditional CMM to generate accurate translations in three orthogonal directions and makes the light pen move with the measuring arm of the CMM. A stationary camera is placed in front of the CMM to capture the images of control points at different positions. For each control point, a separate coordinate system is established, which satisfies that, at each different position, the coordinate of each control point in its respective coordinate system is equal to the reading of the CMM. With the images of control points at different positions and the readings of the CMM, a nonlinear least squares problem is established to optimize the translation vectors between all the control point coordinate systems and camera coordinate system. After a simple conversion, the relative positions of control points can be determined by the differences of these translation vectors. In its essence, the proposed method is an improved version of the SFM method. The moving part changes from the camera to the light pen, and the movement is pure translation that is accurately generated by CMM. Compared with previous SFM method, the proposed method has more known information and far fewer variables to optimize. Each position added to the calibration will give more information about the control point positions and will not introduce any additional optimization variables. Although the proposed method is a vision-based method, the calibration accuracy is more dependent on the accuracy of the CMM. Tens of thousands of translations of the CMM serve as the calibration basis, which allows the proposed method to reach high accuracy. The establishment of control point coordinate systems and the acceleration method of the algorithm guarantee the speed and stability of the solving process.

To be more intuitional, a simple summary of the advantages and disadvantages of the direct measurement method in [10], the structure from motion method in [11], and the proposed calibration method in this paper is presented in Table 1.

Section 2 gives a brief introduction of LPCMS. Section 3 gives a detailed description of the proposed method of control point position calibration. Some experimental results are presented in Section 4 to demonstrate the performance of the proposed method. Section 5 concludes this paper.

## 2. Brief Description of LPCMS

As shown in Figure 1a, a typical LPCMS is composed of a camera, a light pen, a laptop, and some data transmission wires. Figure 1b shows the light pen structure adopted in this paper. It has 13 control points. nine of them are coplanar and form an isosceles triangle. The other four are located in a line that is parallel to the front plane.

In order to describe more clearly, two coordinate systems are established, which are fixed to the camera and the light pen, respectively. The camera coordinate system is denoted as $O_{c}-x_{c}y_{c}z_{c}$, and the light pen coordinate system is denoted as $O_{l}-x_{l}y_{l}z_{l}$. Since the position of the camera is fixed during the measurement, $O_{c}-x_{c}y_{c}z_{c}$ can be regarded as the reference coordinate system. The final measuring result is the coordinate in $O_{c}-x_{c}y_{c}z_{c}$ of the measured point. For any point P, its coordinate in $O_{c}-x_{c}y_{c}z_{c}$ can be denoted as $P_{c}={\left(x_{c},y_{c},z_{c}\right)}^{T}$, and its homogeneous coordinate in $O_{l}-x_{l}y_{l}z_{l}$ can be denoted as ${\tilde{P}}_{l}={\left(x_{l},y_{l},z_{l},1\right)}^{T}$. The coordinates in these two coordinate systems satisfy three dimensional rigid body transformation, which can be denoted as:(1)$$P_{c}=\left[R|T\right]{\tilde{P}}_{l}$$
where R and T are the rotation matrix and translation vector between these two coordinate systems, respectively.

The homogeneous pixel coordinate of its image point can be denoted as $\tilde{p}={\left(u,v,1\right)}^{T}$. According to the pinhole model of camera. The following relationship can be obtained:(2)$$\lambda \tilde{p}=KP_{c}=K\left[R|T\right]{\tilde{P}}_{l}$$
where K is the intrinsic parameter matrix of the camera, and λ is a nonzero scale factor. What needs to be emphasized here is that lens distortion is not considered here, because it is not the main issue of this paper. In practice, distortion compensation is needed to obtain the undistorted pixel coordinate of the image point.

During the measurement, a stationary camera is placed in front of the object to be measured. The operator holds the light pen and guarantees all the control points can be seen by the camera. At the same time, the probe on the light pen is made to touch the surface of the object. The coordinate of the point that the probe touches in $O_{c}-x_{c}y_{c}z_{c}$ can be obtained. LPCMS implements coordinate measurement by analyzing the image of control points on the light pen. With the relative positions of control points and the pixel coordinates of their image points, the rotation matrix R and translation vector T between the camera coordinate system and the light pen coordinate system can be determined. According to Equation (1), the coordinate of the probe tip center in light pen coordinate system $O_{l}-x_{l}y_{l}z_{l}$ can be converted to the coordinate in camera coordinate system $O_{c}-x_{c}y_{c}z_{c}$. As $O_{c}-x_{c}y_{c}z_{c}$ is regarded as the reference coordinate system in LPCMS, the coordinate of the measuring point can be obtained from the coordinate of the probe tip center after a simple radius compensation.

Before actual measurement, LPCMS needs to be carefully calibrated to guarantee accuracy. The system calibration includes camera calibration, control point position calibration, and probe tip center calibration. Camera calibration determines the imaging model of the camera, which includes intrinsic parameters and the distortion coefficients [12,13,14]. In general, the intrinsic parameter matrix is an upper triangular matrix having the following form [15]:(3)$$K=\left[\begin{array}{ccc} f_{x} & s & c_{x}\\ 0 & f_{y} & c_{y}\\ 0 & 0 & 1 \end{array}\right]$$

It uses $f_{x}$ and $f_{y}$ to describe effective focal length in two orthogonal directions of the imaging sensor. s describes the slope of the two axes of the imaging sensor, which results from that the imaging sensor is not strictly vertical to the optical axis. $c_{x}$ and $c_{y}$ are the position of the optical center of the pixel coordinate. Camera calibration is a well-solved problem in computer vision. There is a standard procedure for camera calibration.

Control point position calibration determines the coordinates of control points in the light pen coordinate system $O_{l}-x_{l}y_{l}z_{l}$. Suppose there are n control points for the light pen. Their homogenous coordinates in $O_{l}-x_{l}y_{l}z_{l}$ can be denoted as ${\tilde{P}}_{i}={\left(x_{i},y_{i},z_{i},1\right)}^{T},(i=1,2,\cdots ,n)$. After image processing, the homogenous pixel coordinates of their image points can be denoted as ${\tilde{p}}_{i}={\left(u_{i},v_{i},1\right)}^{T},(i=1,2,\cdots ,n)$. For each control point, two equations can be obtained according to Equation (2) after a simple transformation. A system of 2n equations can be established to calculate the unknown R and T, which represent the rotation matrix and translation vector between camera coordinate system and light pen coordinate system. This is a well-known problem in computer vision, which is called perspective-n-point (PnP) problem. There have been a large number of studies on this issue [16,17,18].

Probe tip center calibration gives the coordinate of the probe tip center in the light pen coordinate system $O_{l}-x_{l}y_{l}z_{l}$, which can be denoted as ${\tilde{P}}_{p}$. With the R and T solved from the PnP problem, the coordinate of the probe tip center in camera coordinate system $O_{c}-x_{c}y_{c}z_{c}$ can be obtained from ${\tilde{P}}_{p}$ after a three-dimensional rigid body transformation according to Equation (1). After compensating for the radius of the probe tip, the coordinate of the measuring point is acquired. Usually, probe tip center can be calibrated using a cone hole [19].

From the description above, control point position calibration is very important to LPCMS. The accuracy of these positions directly impact the solving accuracy of light pen posture R and T, and then impact the accuracy of the final measuring result. However, there are few high-accuracy methods of control point position calibration, which limits the accuracy improvement of LPCMS.

## 3. Control Point Position Calibration

### 3.1. Calibration Procedure

In this paper, a new high accuracy method of control point position calibration, which utilizes a traditional CMM, is proposed. The light pen is fastened to the measuring arm of a CMM with a specially customized fixture, so that there is no relative displacement between the light pen and the measuring arm during the movement of the measuring arm of the CMM. A stationary camera is located in front of the CMM at a proper distance, so that the whole movement range of the CMM is in the view field of the camera. The measuring arm drives the light pen to move along with the pre-programming three-dimensional grid route. At each node of the grid, the image of the light pen is captured by the camera and the reading of the CMM is recorded. For each control point on the light pen, a separate coordinate system is established. The pixel coordinates of the image of the control points can be obtained after image processing. With these image coordinates and the corresponding CMM readings, the translation vectors between all the control point coordinate systems and the camera coordinate system can be calculated. The relative positions of the control points can be defined by the differences of these translation vectors. According to these translation vectors, the light pen coordinate system is established and the coordinates of the control points in the light pen coordinate system can be obtained after a three-dimensional rigid body transformation.

### 3.2. Calibration Method Derivation

As shown in Figure 2, the measuring arm of the CMM drives the light pen to move along with the pre-programming three-dimensional grid route. The machine system of the CMM is denoted as $O_{m}-x_{m}y_{m}z_{m}$ and the probe tip center of the CMM is denoted as point M. The coordinate of point M in the machine system $O_{m}-x_{m}y_{m}z_{m}$ is exactly the reading of the CMM. The control points are denoted as $L_{i},(i=1,2,\cdots ,n)$. For each control point, an individual coordinate system is established, which can be denoted as $O_{i}-x_{i}y_{i}z_{i},(i=1,2,\cdots ,n)$. Their origins are denoted as point $O_{i},(i=1,2,\cdots ,n)$. The control point coordinate system satisfies the condition that, the coordinate of each control point in its own control point coordinate system is equal to the reading of the CMM, at each position of the moving route. If the control point coordinate systems are established like this, then the three axes of the control point coordinate systems have the same direction as the homonymous axes of the machine coordinate system. In other words, the difference between any two coordinate systems is only a pure translation.

The detail derivation process can be formulized as follows. The condition that control point coordinate system satisfies can be expressed in vector form:(4)$$\vec{O_{m}M}=\vec{O_{i}L_{i}}$$

Then the four points $O_{m},M,O_{i},L_{i}$ form a parallelogram. Thus, the following relation is also satisfied:(5)$$\vec{O_{m}O_{i}}=\vec{ΜL_{i}},(i=1,2,\cdot \cdot \cdot ,n)$$

Take the first and second control point for example, the following relation can be derived:(6)$$\begin{array}{c} \begin{array}{l} \vec{O_{m}O_{1}}=\vec{ML_{1}},\vec{O_{m}O_{2}}=\vec{ML_{2}} \end{array}\\ \begin{array}{l} \vec{O_{m}O_{2}}-\vec{O_{m}O_{1}}=\vec{ML_{2}}-\vec{ML_{1}} \end{array}\\ \vec{O_{1}O_{2}}=\vec{L_{1}L_{2}} \end{array}$$

From Equation (6), it can be concluded that the relative positions of the origins of the control point coordinate systems $O_{i},(i=1,2,\cdots ,n)$ is the same as the relative positions of the control points $L_{i},(i=1,2,\cdots ,n)$. Then the control point position calibration problem is converted to determining the relative positions of the origins of control point coordinate systems.

As shown in Figure 3, for each control point coordinate system $O_{i}-x_{i}y_{i}z_{i},(i=1,2,\cdots ,n)$, its relationship relative to the camera coordinate system can be represented by a three-dimensional rigid body transformation. The translation vectors are denoted as $T_{i},(i=1,2,\cdots ,n)$. The physical meaning of $T_{i}$ is the vector form the origin of camera coordinate system to the origin of the control point coordinate system, which can be formulated as:(7)$$\vec{O_{c}O_{i}}=T_{i},(i=1,2,\cdots ,n)$$

Take the first and second control points for example, the following relationship can be derived:(8)$${\begin{array}{c} \begin{array}{l} \vec{O_{c}O_{1}}=T_{1},\vec{O_{c}O_{2}}=T_{2} \end{array}\\ \vec{O_{1}O_{2}}=\vec{O_{c}O_{2}}-\vec{O_{c}O_{1}}=T_{2}-T \end{array}}_{1}$$

From Equation (8), it can be concluded that the relative positions of the origins of control point coordinate systems can be determined by calculating the differences of a series of translation vectors, which represent the translation between the control point coordinate systems and the camera coordinate system.

From the derivation above, the calibration of the control point positions is transformed to calculating the translation vector between the control point coordinate systems and the camera coordinate system. If each control point is processed separately, this calculation can be regarded as an instance of the PnP problem, and the rotation matrix between control point coordinate system and camera coordinate system is calculated at the same time. Suppose there are m positions on the three dimensional grid route of the CMM. The homogeneous form of the readings of the CMM can be denoted as ${\tilde{Q}}_{j}={\left(x_{j},y_{j},z_{j},1\right)}^{T},(j=1,2,\cdots ,m)$. According to the establishment condition of the control point coordinate systems mentioned above, ${\tilde{Q}}_{j}={\left(x_{j},y_{j},z_{j},1\right)}^{T},(j=1,2,\cdots ,m)$ are also the coordinates of a control point in its own control point coordinate system. After image processing, the homogeneous pixel coordinates of the image of control points can be denoted as ${\tilde{q}}_{i,j}={\left(u_{i,j},v_{i,j},1\right)}^{T},(i=1,2,\cdots ,n;j=1,2,\cdots ,m)$. The rotation matrixes and translation vectors between control point coordinate systems and camera coordinate systems can be denoted as $R_{i},(i=1,2,\cdots ,n)$ and $T_{i}={\left(t_{x,i},t_{y,i},t_{z,i}\right)}^{T},(i=1,2,\cdots ,n)$, respectively.

Take the i-th control point for example: The following equations can be derived from Equation (2) at each position of the route:(9)$$\lambda \left[\begin{array}{c} u_{i,j}\\ v_{i,j}\\ 1 \end{array}\right]=\left[\begin{array}{ccc} f_{x} & s & c_{x}\\ 0 & f_{y} & c_{y}\\ 0 & 0 & 1 \end{array}\right]\left[R_{i}|T_{i}\right]\left[\begin{array}{c} x_{j}\\ y_{j}\\ z_{j}\\ 1 \end{array}\right],(j=1,2,\cdots ,m)$$
where $R_{i}$ is a 3×3 orthogonal matrix. An orthogonal matrix has only three degrees of freedom, and there are several methods to parameterize it. In this paper, rotation angles around coordinate axes, which can be denoted as $\left(\alpha_{i},\beta_{i},\gamma_{i}),(i=1,2,\cdots ,n\right)$, are used to parameterize the rotation matrix as follows:(10)$$R_{i}=\left[\begin{array}{ccc} \begin{array}{c} \cos\beta_{i}\cos\gamma_{i}\\ \cos\beta_{i}\sin\gamma_{i}\\ \cos\gamma_{i} \end{array} & \begin{array}{c} \sin\alpha_{i}\sin\beta_{i}\cos\gamma_{i}-\cos\alpha_{i}\sin\gamma_{i}\\ \sin\alpha_{i}\sin\beta_{i}\sin\gamma_{i}+\cos\alpha_{i}\cos\gamma_{i}\\ \sin\alpha_{i}\cos\beta_{i} \end{array} & \begin{array}{c} \cos\alpha_{i}\sin\beta_{i}\cos\gamma_{i}+\sin\alpha_{i}\sin\gamma_{i}\\ \cos\alpha_{i}\sin\beta_{i}\sin\gamma_{i}-\sin\alpha_{i}\cos\gamma_{i}\\ \cos\alpha_{i}\cos\beta_{i} \end{array} \end{array}\right]$$

What needs to be emphasized here is that the rotation matrix has several different denotations parameterized by rotation angles around the coordinate axes, depending on the order of three rotations. The denotation in Equation (10) is only one of them. Expanding Equation (9), the scale factor λ can be eliminated. An equation system composed of 2m nonlinear equations with six unknown variables $\left(\alpha_{1},\beta_{i},\gamma_{i},t_{x,i},t_{y,i},t_{z,i}\right)$ is established. Because the detailed form of the equations is too complicated, the following abbreviated form is used in this paper:(11)$$\{\begin{array}{cc} \begin{array}{c} \begin{array}{l} g_{i,j}\left(\alpha_{i},\beta_{i},\gamma_{i},t_{x,i},t_{y,i},t_{z,i}\right)-u_{i,j}=0 \end{array}\\ h_{i,j}\left(\alpha_{i},\beta_{i},\gamma_{i},t_{x,i},t_{y,i},t_{z,i}\right)-v_{i,j}=0 \end{array} & \left(j=1,2,\cdots ,m\right) \end{array}$$

This is a general nonlinear least squares problem, and there are many mature solutions to this problem. Then the translation vector between the i-th control point coordinate system and camera coordinate system $T_{i}={\left(t_{x,i},t_{y,i},t_{z,i}\right)}^{T}$ is determined.

For each control point, the same process is repeated, and the only change is to use different image coordinates. Then all the translation vectors $T_{i}={\left(t_{x,i},t_{y,i},t_{z,i}\right)}^{T},(i=1,2,\cdots ,n)$ are determined. According to the derivation in Equations (6) and (8), the relative positions of the control points can be calculated from the differences of these translation vectors.

### 3.3. Algorithm Acceleration

At the beginning of Section 3.2, it was mentioned that the way that control point coordinate systems are established guarantees that the difference between any two control point coordinate systems is a pure translation. Thus, their rotation matrices relative to the camera coordinate system are the same, which can be represented by the following formula:(12)$$\begin{array}{cc} \left\{ \begin{array}{l} \alpha_{i1}=\alpha_{i2}=\alpha \\ \beta_{i1}=\beta_{i2}=\beta \\ \gamma_{i1}=\gamma_{i2}=\gamma \end{array}\right. & \left(i1,i2=1,2,\cdots ,n;i1\neq i2\right) \end{array}$$

With these conditions satisfied, a better way to calculate the translation vectors is to take all control points into consideration at the same time rather than calculate them separately. If all control points are handled at the same time, two equations can be derived from Equation (2) for each control point, at each position of the CMM route. The equation system in Equation (11) for all the control point can be merged together. As a result, an equation system which is composed of 2mn equations with 3n+3 variables $\left(\alpha_{i},\beta_{i},\gamma_{i},t_{x,i},t_{y,i},t_{z,i}),(i=1,2,\cdots ,n\right)$ is established, which has a similar form to Equation (11):(13)$$\{\begin{array}{cc} \begin{array}{c} \begin{array}{l} g_{i,j}\left(\alpha ,\beta ,\gamma ,t_{x,i},t_{y,i},t_{z,i}\right)-u_{i,j}=0 \end{array}\\ h_{i,j}\left(\alpha ,\beta ,\gamma ,t_{x,i},t_{y,i},t_{z,i}\right)-v_{i,j}=0 \end{array} & \left(i=1,2,\cdots ,n;j=1,2,\cdots ,m\right) \end{array}$$

This equation system can be solved as a general nonlinear least squares problem. In computer vision, the solving process is also known as the bundle adjustment method [20,21].

In practical calibration process, there are tens of thousands different positions on the CMM route to guarantee the calibration accuracy, so m is a quite large number. Take the light pen in Figure 1b for example, n=13 and there are totally 42 variables in the equation system. This means the equation system in Equation (13) will be very large. Solving it directly will be time and space consuming.

Take the classical Gauss–Newton method with line search for example [22]. It starts from an initial solution vector, and uses a gradient-based iteration framework to approximate the locally optimal solution. Suppose X represents the solution vector of the system. The initial solution vector can be denoted as $X^{\left(0\right)}=\left(\alpha^{\left(0\right)},\beta^{\left(0\right)},\gamma^{\left(0\right)},t_{x,i}^{\left(0\right)},t_{y,i}^{\left(0\right)},t_{z,i}^{\left(0\right)}\right),(i=1,2,\cdots ,n)$ and the solution vector in the k-th iteration can be denoted as $X^{\left(k\right)}=\left(\alpha^{\left(k\right)},\beta^{\left(k\right)},\gamma^{\left(k\right)},t_{x,i}^{\left(k\right)},t_{y,i}^{\left(k\right)},t_{z,i}^{\left(k\right)}\right),(i=1,2,\cdots ,n)$. The iteration criterion is as follows:(14)$$X^{\left(k+1\right)}=X^{\left(k\right)}-d^{\left(k\right)}Z^{\left(k\right)}$$
where $Z^{\left(k\right)}$ is the iteration direction at $X^{\left(k\right)}$ and $d^{\left(k\right)}$ is the step length along $Z^{\left(k\right)}$. $Z^{\left(k\right)}$ is the solution of a linear equation system. It can be calculated through the following formula:(15)$$Z^{\left(k\right)}={\left(J^{{\left(k\right)}^{T}}J^{\left(k\right)}\right)}^{-1}J^{{\left(k\right)}^{T}}R^{\left(k\right)}$$
where $J^{\left(k\right)}$ and $R^{\left(k\right)}$ are the Jacobian matrix and residual vector of the equations system at $X^{\left(k\right)}$, respectively. The entry of $J^{\left(k\right)}$ is the first-order partial derivative of the equations with respect to all variables. The value of $d^{\left(k\right)}$ can be calculated through one-dimensional line search along $Z^{\left(k\right)}$. For the equation system in Equation (13), $J^{\left(k\right)}$ is a 2mn×(3n+3) matrix, which is very large, and $J^{{\left(k\right)}^{T}}J^{\left(k\right)}$ is a (3n+3)×(3n+3) matrix. The storage of $J^{\left(k\right)}$ will occupy quite a large amount of memory and the computation of $J^{{\left(k\right)}^{T}}J^{\left(k\right)}$ will consume a great deal of time. This computation will be conducted once during each iteration, so special treatment is needed to accelerate the algorithm.

Although there are 3n+3 variables in the whole equation system, any single equation has only six variables, including three rotation angles and three translation components. The 2m equations of one control point have nothing to do with the translation vectors of the other control points. Therefore, the following partial derivatives are zero:(16)$$\begin{array}{cc} \begin{array}{c} \begin{array}{l} \frac{\partial h_{i1,j}}{\partial t_{x,i2}}=\frac{\partial h_{i1,j}}{\partial t_{y,i2}}=\frac{\partial h_{i1,j}}{\partial t_{z,i2}}=0 \end{array}\\ \frac{\partial g_{i1,j}}{\partial t_{x,i2}}=\frac{\partial g_{i1,j}}{\partial t_{y,i2}}=\frac{\partial g_{i1,j}}{\partial t_{z,i2}}=0 \end{array} & \begin{array}{c} \left(i1,i2=1,2,\cdots ,n;i1\neq i2\right)\\ \left(j=1,2,\cdots ,m\right) \end{array} \end{array}$$

In other words, the Jacobian matrix J is a sparse matrix, whose most entries are zero. The algorithm can be accelerated by taking advantage of the sparsity of J [23,24]. The Jacobian matrix J can be organized in partitioned form as:(17)$$J=\left[\begin{array}{cc} \begin{array}{c} \begin{array}{l} A_{1} \end{array}\\ A_{2}\\ \vdots \\ A_{n} \end{array} & \begin{array}{cccc} \begin{array}{l} B_{1} \end{array} & \begin{array}{l} 0 \end{array} & \begin{array}{l} \cdots \end{array} & \begin{array}{l} 0 \end{array}\\ 0 & B_{2} & \cdots & 0\\ \vdots & \vdots & \ddots & \vdots \\ 0 & 0 & \cdots & B_{n} \end{array} \end{array}\right]$$
where $A_{i},(i=1,2,\cdots ,n)$ is a 2m×3 matrix composed of partial derivatives with respect to rotation angles (α,β,γ), and $B_{i},(i=1,2,\cdots ,n)$ is a 2m×3 matrix composed of partial derivatives with respect to translation components $\left(t_{xi},t_{yi},t_{zi}),(i=1,2,\cdots ,n\right)$. They have the following form:(18)$$A_{i}=\left[\begin{array}{c} \begin{array}{ccc} \begin{array}{l} \frac{\partial h_{i,1}}{\partial \alpha } \end{array} & \begin{array}{l} \frac{\partial h_{i,1}}{\partial \beta } \end{array} & \begin{array}{l} \frac{\partial h_{i,1}}{\partial \gamma } \end{array}\\ \frac{\partial g_{i,1}}{\partial \alpha } & \frac{\partial g_{i,2}}{\partial \beta } & \frac{\partial g_{i,3}}{\partial \gamma } \end{array}\\ \begin{array}{l} \vdots \end{array}\\ \begin{array}{ccc} \begin{array}{l} \frac{\partial h_{i,m}}{\partial \alpha } \end{array} & \begin{array}{l} \frac{\partial h_{i,m}}{\partial \beta } \end{array} & \begin{array}{l} \frac{\partial h_{i,m}}{\partial \gamma } \end{array}\\ \frac{\partial g_{i,m}}{\partial \alpha } & \frac{\partial g_{i,m}}{\partial \beta } & \frac{\partial g_{i,m}}{\partial \gamma } \end{array} \end{array}\right],B_{i}=\left[\begin{array}{c} \begin{array}{ccc} \begin{array}{l} \frac{\partial h_{i,1}}{\partial t_{x,i}} \end{array} & \begin{array}{l} \frac{\partial h_{i,1}}{\partial t_{y,i}} \end{array} & \begin{array}{l} \frac{\partial h_{i,1}}{\partial t_{z,i}} \end{array}\\ \frac{\partial g_{i,1}}{\partial t_{x,i}} & \frac{\partial g_{i,1}}{\partial t_{y,i}} & \frac{\partial g_{i,1}}{\partial t_{z,i}} \end{array}\\ \begin{array}{l} \vdots \end{array}\\ \begin{array}{ccc} \begin{array}{l} \frac{\partial h_{i,m}}{\partial t_{xi}} \end{array} & \begin{array}{l} \frac{\partial h_{i,m}}{\partial t_{yi}} \end{array} & \begin{array}{l} \frac{\partial h_{i,m}}{\partial t_{zi}} \end{array}\\ \frac{\partial g_{i,m}}{\partial t_{x,i}} & \frac{\partial g_{i,m}}{\partial t_{y,i}} & \frac{\partial g_{i,m}}{\partial t_{z,i}} \end{array} \end{array}\right]$$

Then $J^{T}J$ can be computed in partitioned form:(19)$$J^{T}J=\left[\begin{array}{cccc} \sum_{i=1}^{n}A_{i}^{T}A_{i} & A_{1}^{T}B_{1} & \cdots & A_{n}^{T}B_{n}\\ B_{1}^{T}A_{1} & B_{1}^{T}B_{1} & \cdots & 0\\ \vdots & \vdots & \ddots & \vdots \\ B_{n}^{T}A_{n} & 0 & \cdots & B_{n}^{T}B_{n} \end{array}\right]$$

From Equation (19), it can be seen that, with the acceleration method, only one $A_{i}$ and one $B_{i}$ are stored at any moment of the algorithm. Once the 3×3 subblocks $A_{i}^{T}A_{i},A_{i}^{T}B_{i},B_{i}^{T}B_{i}$ are calculated, the values of $A_{i}$ and $B_{i}$ can be updated. Therefore, the memory occupied by Jacobian matrix J reduces from 2mn×(3n+3) to 2m×6. There are 3n subblocks to calculate in total, and the time complexity of one calculation is $2m\times 3^{2}$. Therefore, the time complexity to calculate $J^{T}J$ reduces from $2mn\times {\left(3n+3\right)}^{2}$ to $3n\times 2m\times 3^{2}$. It can be seen that the proposed method above, which utilizes the sparsity of Jacobian matrix, can accelerate the solving algorithm of Equation (13) significantly.

### 3.4. Light Pen Coordinate System Establishement

After solving the equation system in Equation (13), the translation vectors $T_{i}={\left(t_{x,i},t_{y,i},t_{z,i}\right)}^{T},(i=1,2,\cdots ,n)$ between the control point coordinate systems and camera coordinate system are obtained. According to the derivation in Equations (8) and (10), the relative positions of the control points can be determined by the differences of these translation vectors. However, these relative positions are represented with the coordinates in the camera coordinate system, which vary with the position of the camera during the calibration process. For convenience, there is a need to establish the light pen coordinate system $O_{l}-x_{l}y_{l}z_{l}$ according to the structure of the light pen. The light pen coordinate system can be established in many ways. For the light pen in Figure 1b, the light pen coordinate system can be established as follows:

Determine the origin of $O_{l}-x_{l}y_{l}z_{l}$. The first control point is used as the origin. Then a new set of translation vectors $T_{i}^{\prime }={\left(t_{x,i}{}^{\prime },t_{y,i}{}^{\prime },t_{z,i}{}^{\prime }\right)}^{T},(i=1,2,\cdots ,13)$ is calculated:(20)$$\begin{array}{cc} \{\begin{array}{c} t_{x,i}{}^{\prime }=t_{x,i}-t_{x,1}\\ t_{y,i}{}^{\prime }=t_{y,i}-t_{y,1}\\ t_{z,i}{}^{\prime }=t_{z,i}-t_{z,1} \end{array} & \left(i=1,2,\cdots ,13\right) \end{array}$$Determine the z-axis of $O_{l}-x_{l}y_{l}z_{l}$. The fifth to 13th control points are designed to be coplanar. Actually, they are not strictly coplanar because of the machining and installation error. Therefore, a plane is fitted using their new translation vectors $T_{i}^{\prime }={\left(t_{x,i}{}^{\prime },t_{y,i}{}^{\prime },t_{z,i}{}^{\prime }\right)}^{T},(i=5,6,\cdots ,13)$. The direction of its unit normal vector, which can be denoted as q, is used as the z-axis of $O_{l}-x_{l}y_{l}z_{l}$.Determine the y-axis of $O_{l}-x_{l}y_{l}z_{l}$. For the same reason, the first to fourth control points are not strictly collinear. They are projected to the plane fitted before. A line is fitted using the coordinates of their projections. The unit direction vector of this line is denoted as p. Since the fitted line is in the plane, p⊥q is satisfied. The direction of p can be used as the y-axis of $O_{l}-x_{l}y_{l}z_{l}$.Determining the x-axis of $O_{l}-x_{l}y_{l}z_{l}$. Given the direction of y-axis p and z-axis q, the direction of x-axis, which is denoted as o, can be determined by the cross product:(21)o=p×qDetermining the relative positions of control points with coordinates in $O_{l}-x_{l}y_{l}z_{l}$. With the origin and axes determined, the coordinates of control points in $O_{l}-x_{l}y_{l}z_{l}$, which can be denoted as $P_{i},(i=1,2,\cdots ,13)$ can be calculates as follows:(22)$$P_{i}=\left[\begin{array}{c} o^{T}\\ p^{T}\\ q^{T} \end{array}\right]T_{i}^{\prime },(i=1,2,\cdots ,13)$$

## 4. Experiment

### 4.1. Actual Control Point Position Calibration Experiment

Figure 4a shows the scene of one actual experiment, and Figure 4b shows how the light pen is fastened to the measuring arm with the special customized fixture.

The light pen is made of carbon fiber reinforced materials and manufactured by the use of a customized mold, which is designed according to the light pen structure shown in Figure 1b. The control points are infrared LEDs, which are assembled to the light pen by a screw joint. There is also a small control board on the light pen, which can receive the instruction from the laptop. The luminance of the LEDs can be adjusted as required. The CMM is Global Silver Classic SR supplied by Hexagon, Qingdao, Shandong Province, China. Its positioning accuracy is 2.3 μm. The camera is an industrial camera with 5 million pixels, which is supplied by Ximea, Münster, North Rhine-Westphalia, Germany. The lens is an industrial lens with 16 mm focal length, which is supplied by Satoo, Saitama, Japan. The camera is located in front of the CMM at a proper distance, so that the light pen is always in the view field of the camera during the calibration process. Take the calibration experiment in Figure 4a for example: the camera is located about 1.5 m in front of the CMM. The movement ranges in three directions are all 400 mm, which is limited within the movement range of the CMM. The three-dimensional grid route has m=20 × 20 × 20=8000 nodes and, therefore, the distance between two adjacent nodes is 20 mm. It takes about 2.5 h to finish the calibration.

In order to eliminate the influence of room ambient light, two approaches are adopted in the calibration procedure as well as the actual measurement process. First, the control points are infrared LEDs, and an infrared light filter is installed in front of the camera lens. Then the visible light component of the room ambient light is filtered out. Second, the frame that is used for image processing is not a frame directly captured by the camera. It is obtained by calculating the difference between two adjacent frames, one is the frame that the LEDs are lighted and the other is a frame that only contains the background. These two approaches will reduce the influence of room ambient light to a large extent, if the environmental light condition does not change rapidly.

The accuracy of the translational movement of the light pen is guaranteed by the CMM. Another factor that affects the accuracy of the calibration is the accuracy of the pixel coordinates of the control points’ images. To achieve high accuracy, an image processing method with sub-pixel accuracy is required. For the reason that the energy distribution of the infrared LEDs on the imaging sensor is approximate to a two-dimensional Gaussian distribution, the center of the image point can be obtained by Gaussian surface fitting from the gray values of the feature area. In practice, the repeatability of the pixel coordinate is 0.05 pixels.

With the proposed method, the rotation angles between control point coordinate systems and camera coordinate system are α=91.717°, β=0.782°, γ=−1.255°. The translation vectors $T_{i}={\left(t_{x,i},t_{y,i},t_{z,i}\right)}^{T},(i=1,2,\cdots ,13)$ are listed in Table 2.

Using the method in Section 3.4 to establish the light pen coordinate system, the direction of the three axes are:$$o={\left(-0.01445,-0.9998,-0.01335\right)}^{T}$$
$$p={\left(-0.9997,0.01421,0.01883\right)}^{T}$$
$$q={\left(-0.01864,0.01362,-0.9997\right)}^{T}$$

Then the control point positions in the light pen coordinate system $P_{i}={\left(x_{i},y_{i},z_{i}\right)}^{T},(i=1,2,\cdots ,13)$, which are the final result of this calibration, can be calculated from Equations (20) and (22). The results are listed in Table 3.

To illustrate the repeatability of the proposed calibration method, 10 calibrations are conducted by locating the camera at different positions. Due to the existence of error, the results of 10 calibrations are slightly different. Therefore, the light pen coordinate systems established in these calibrations are also slightly different. It is more reasonable to use the distances of any two control points, rather than the coordinates in the light pen coordinate system, to measure the repeatability of the proposed method. For 13 control points, there are 78 distances to calculate in total. Owing to space constraints, only seven representative distances are listed in Table 4 to illustrate the repeatability. From Table 4, it can be concluded the repeatability of calibration with the proposed method is about 10 μm.

### 4.2. Simulation Experiment

Since the true values of control point positions are unknown, actual calibration experiments can only reflect the repeatability of the proposed method. As to accuracy, we can only resort to simulation experiments. The procedure of the simulation experiment is performed as follows: Given a set of control point positions as the true value, the rigid body transformation between the CMM and the camera and the readings at each node of the CMM route are also given. Equations (9), (20) and (22) are used to trace back to obtain the ideal pixel coordinates of the image points. To verify the accuracy of the proposed method, random errors of different ranges that obey the normal distribution are added to the readings of the CMM and the ideal image coordinates to simulate the actual situation. With the disturbed data, calibration calculation is performed to obtain the simulation result. The performance of the proposed method can be evaluated by comparing the calibration result and the given true value. In addition to random errors, the influence of the number of nodes on calibration accuracy is also studied. Figure 5a shows the calibration errors when different ranges of random errors, which obey the normal distribution, are added to the CMM readings and image coordinates. The deviation sequences of two random errors adopted in the simulation experiment are [1.0,2.0,3.0,4.0,5.0] μm and [0.01,0.02,0.05,0.1,0.2] pixels, respectively. At the same time, the number of nodes on the CMM route is fixed at m=30 × 30 × 30=27,000. Figure 5b shows the calibration errors with different numbers of nodes on the CMM route, when the deviation standards of two random errors are fixed at 0.1 pixel and 4 μm, respectively. The sequence of node numbers is [1000,8000,27000,64000,125000]. The movement ranges in three directions are all fixed at 400 mm and, therefore, the distance sequence of two adjacent nodes is [40,20,13.333,10,8] mm.

From Figure 6a, it can be seen that the simulated calibration error is approximately proportional to the deviation of the random error added to the CMM readings and image coordinates. From Figure 6b, it can be seen that when the number of nodes is small, the calibration error decreases significantly with the increase of the node number. However, when the node number exceeds a certain number, the calibration error is almost unchanged. Under this circumstance, the increase in the number of nodes takes a great deal of time, but provides little benefit to the calibration.

### 4.3. Measurement Experiment of LPCMS

With the calibration results in Table 3, LPCMS is used to measure three standard gauges and a standard cylindrical bore on a standard module, which are shown in Figure 6, 10 times at distances from 2 to 10 m. The nominal lengths of the gauges are 100 mm, 250 mm, and 1000 mm, respectively, and the nominal diameter of the cylindrical bore is 2.5 inches, i.e., 63.5 mm. What needs to be explained here is that, once the relative positions of the control points are determined, the light pen can translate and rotate arbitrarily as long as it can be seen in the view of field of the camera. The light pen is held by hand during the measuring process and, therefore, not only translation but also rotation inevitably occurs in the measurement. For each measured point, the current posture of the light pen is calculated by solving the PnP problem. Owing to space constraints, only the statistical results are shown in Table 5. From the results in Table 5, it can be concluded that the standard deviation of the measurement of the whole LPCMS can achieve 30 μm within 10 m.

To compare the performance of the existing and proposed calibration methods quantitatively, three different groups of control point positions, which are calibrated with the methods in [10,11] and the proposed method in this paper, respectively, are used in the LPCMS to measure the 250 mm gauge at different distances from 2 to 10 m. The statistical results are shown in Table 6. From Table 6, it can be seen that the proposed method is superior to existing methods in terms of accuracy.

## 5. Conclusions

In this paper, a high-accuracy CMM-based method of control point position calibration for LPCMS is proposed. A traditional CMM is used to drive the light pen to move along a three dimensional grid route. Through establishing a coordinate system for each control point in a certain way, the relative positions of control points can be transformed to the differences of the translation vectors between control point coordinate systems and camera coordinate system. A nonlinear least squares problem is established to solve these translation vectors. Since the number of variables is only relative to the number of control points, there can be tens of thousands of nodes on the CMM route so that the influence of random error is significantly reduced. The positional accuracy of each node is guaranteed by the CMM, so the CMM can be seen as the calibration basis of the proposed method. Since the nonlinear least squares problem above is quite sparse, a method to accelerate the algorithm is also proposed, which makes it possible to handle the vast number of nodes.

Although the proposed method is a vision-based method, the calibration accuracy is more dependent on the accuracy of the CMM. Therefore, compared with existing vision-based methods, the proposed method is more accurate. The establishment of control point coordinate systems and the acceleration method of the algorithm guarantees the speed and stability of the solving process. Experiments show that, with the proposed calibration method, the whole LPCMS can reach a rather high accuracy.

## Figures and Tables

**Figure 1 sensors-20-05592-f001:**
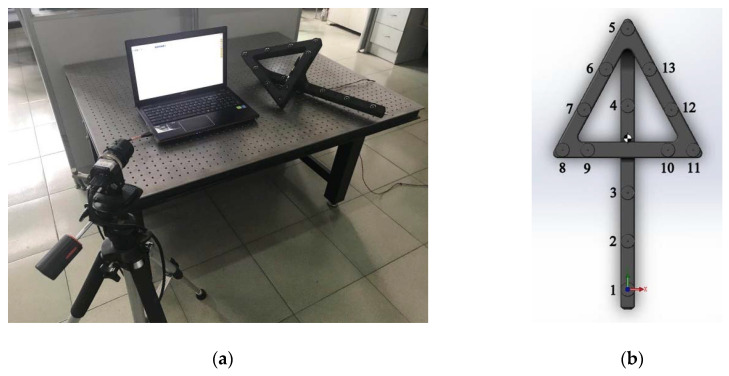
Introduction of light pen coordinate measuring system. (**a**) The typical composition of the light pen coordinate measuring system; (**b**) the light pen structure adopted in this paper.

**Figure 2 sensors-20-05592-f002:**
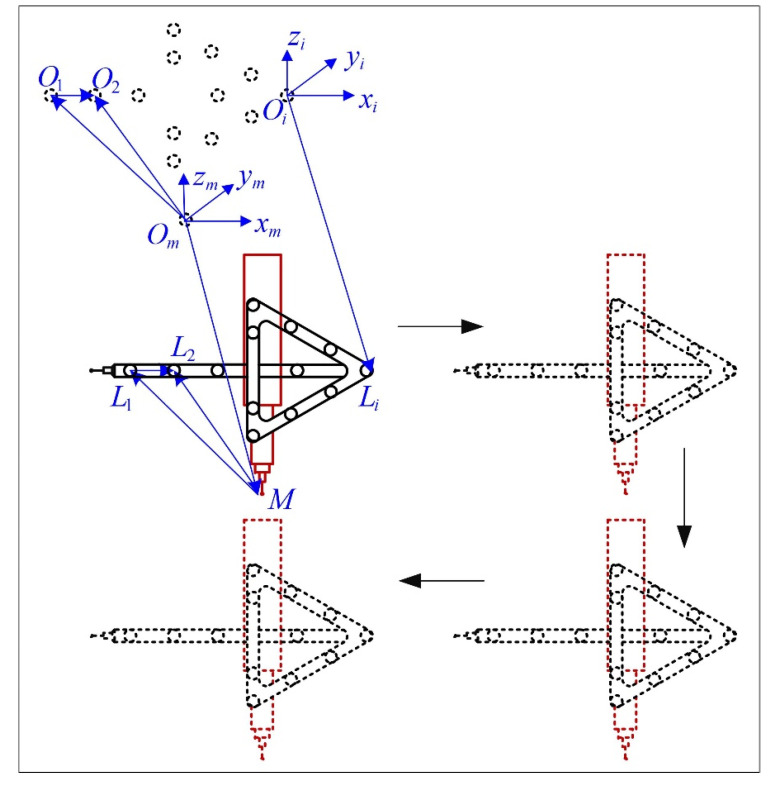
The establishment of the CMM machine coordinate system and all the control point coordinate systems. The control point position calibration problem is converted to determining the relative positions of the origins of control point coordinate systems.

**Figure 3 sensors-20-05592-f003:**
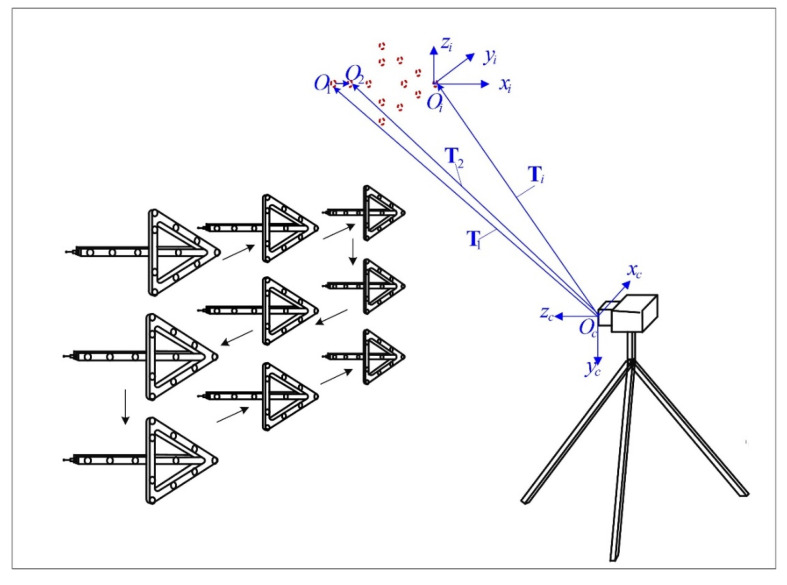
The relationship between the control point coordinate systems and the camera coordinate system. Determining the relative positions of the origins of control point coordinate systems can be converted to calculating the translation vectors between the control point coordinate systems and the camera coordinate systems.

**Figure 4 sensors-20-05592-f004:**
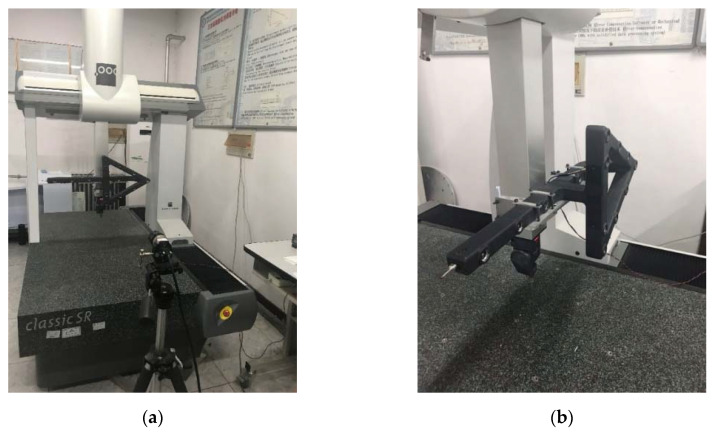
Actual experiment of control point position calibration: (**a**) The scene of one actual experiment; (**b**) the customized fixture to fasten light pen.

**Figure 5 sensors-20-05592-f005:**
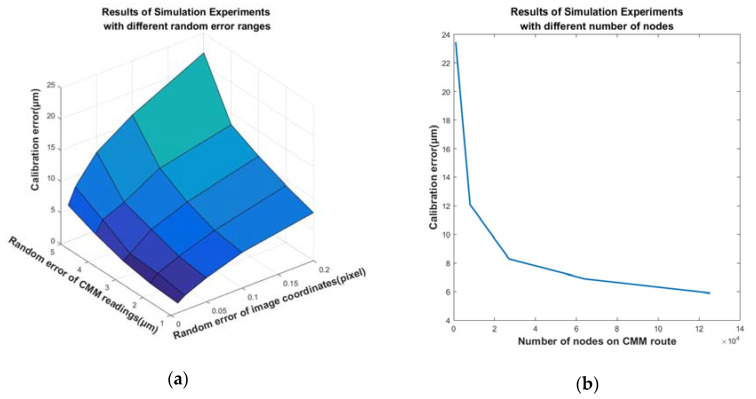
Calibration results of simulation experiments: (**a**) Calibration errors with different ranges of random errors. The random errors obey normal distribution, and the scale values are different standard deviations. (**b**) Calibration errors with different numbers of nodes on the CMM route.

**Figure 6 sensors-20-05592-f006:**
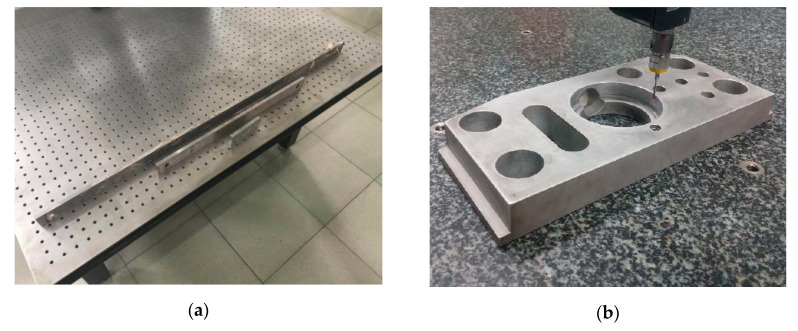
Measurement experiments of LPCMS: (**a**) Three gauges of different lengths; (**b**) a standard cylindrical bore.

**Table 1 sensors-20-05592-t001:** Comparison table of the pros and cons of calibration methods.

Calibration Method	Advantages	Disadvantages
Direct Measurement Method in [10]	Fast calibration speed; Easy operation; No need of calculation	Unable to locate luminous center; Low accuracy
Structure from Motion Method in [11]	No need of extra equipment; High flexibility; Able to calibrate in real-time	Complicated calculation; Low reliability
The Proposed Method	High accuracy; Simple calculation process; Able to process tremendous data	High time consumption; Complex calibration procedure

**Table 2 sensors-20-05592-t002:** Translation vectors between control point coordinate systems and the camera coordinate system at the position in Figure 4a (mm).

Control Point Index	$t_{x,i}$	$t_{y,i}$	$t_{z,i}$
1	−31.343	−319.578	1542.853
2	−130.327	−317.944	1544.784
3	−230.340	−316.726	1546.672
4	−410.261	−314.924	1550.004
5	−571.145	−309.694	1452.690
6	−486.863	−266.087	1451.879
7	−402.882	−222.740	1451.028
8	−318.265	−179.508	1450.131
9	−319.429	−234.657	1449.130
10	−322.394	−394.058	1447.287
11	−323.049	−449.060	1446.452
12	−405.913	−402.371	1448.562
13	−488.814	−355.599	1450.688

**Table 3 sensors-20-05592-t003:** Control point positions in the light pen coordinate system (mm).

Control Point Index	$x_{i}$	$y_{i}$	$z_{i}$
1	0.000	0.000	0.000
2	−0.228	99.159	−0.064
3	−0.025	199.053	−0.071
4	0.730	379.013	−0.024
5	−0.874	538.094	100.333
6	−45.681	454.440	100.167
7	−90.222	371.082	100.043
8	−134.657	287.086	99.951
9	−79.488	287.447	100.223
10	79.949	288.113	99.950
11	134.962	287.970	100.048
12	89.451	371.514	100.119
13	43.858	455.096	100.176

**Table 4 sensors-20-05592-t004:** Repeatability of nine representative distances between two control points. **Ave** is the abbreviation of average value. **Std** is the abbreviation of standard deviation. (mm).

Calibration Index	1–4 ^1^	1–5	5–8	5–11	8–11	1–8	1–11
1	379.014	547.369	284.435	284.629	269.620	332.471	333.393
2	379.022	547.385	284.446	284.611	269.599	332.476	333.425
3	379.012	547.355	284.461	284.603	269.625	332.483	333.420
4	379.030	547.344	284.447	284.622	269.615	332.450	333.409
5	379.005	547.363	284.452	284.631	269.603	332.456	333.396
6	379.024	547.359	284.458	284.615	269.621	332.464	333.433
7	379.019	547.388	284.432	284.623	269.608	332.469	333.401
8	379.008	547.354	284.441	284.618	269.619	332.472	333.422
9	379.027	547.376	284.459	284.607	269.618	332.456	333.411
10	379.016	547.371	284.429	284.614	269.603	332.448	333.430
**Ave**	379.018	547.366	284.446	284.617	269.613	332.465	333.414
**Std**	0.00818	0.01408	0.01158	0.00911	0.00909	0.01165	0.01417
**Range**	0.025	0.044	0.032	0.028	0.026	0.035	0.040

^1^ The denotation i-j means the distance between the i-th and j-th control points.

**Table 5 sensors-20-05592-t005:** The results of measurement experiment of LPCMS. The meanings of the abbreviations **Ave** and **Std** are the same as in Table 4 (mm).

Distance	100 Gauge		250 Gauge		1000 Gauge		Cylinder	
	Ave	Std	Ave	Std	Ave	Std	Ave	Std
2 m	99.994	0.0025	250.009	0.0028	999.982	0.0062	63.502	0.0022
4 m	100.002	0.0037	250.014	0.0049	999.994	0.0133	63.507	0.0027
6 m	100.006	0.0048	249.983	0.0072	1000.015	0.0187	63.513	0.0042
8 m	99.988	0.0056	250.003	0.0123	999.989	0.0227	63.505	0.0053
10 m	99.992	0.0071	250.007	0.0158	999.979	0.0315	63.519	0.0085

**Table 6 sensors-20-05592-t006:** Measurement results of 250 mm gauge with different calibration results. The meanings of the abbreviations **Ave** and **Std** are the same as in Table 4 (mm).

Distance	Method in [10]		Method in [11]		Proposed Method	
	Ave	Std	Ave	Std	Ave	Std
2 m	250.033	0.0258	250.019	0.0121	250.009	0.0028
4 m	249.976	0.0376	250.033	0.0213	250.014	0.0049
6 m	249.982	0.0627	249.975	0.0381	249.983	0.0072
8 m	250.041	0.1032	250.029	0.0419	250.003	0.0123
10 m	249.963	0.1523	250.038	0.0688	250.007	0.0158

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
