# Peer review of "A CMM-Based Method of Control Point Position Calibration for Light Pen Coordinate Measuring System"

_sensors, 2020, doi:10.3390/s20195592_

Round 1

Reviewer 1 Report

Comments:

This paper describes a CMM-based method of control point position calibration for light pen coordinate measuring system (LPCMS). The paper has certain novelty and advantages for this field research work, and has value for publishing in Journal of Sensors.

I suggest this manuscript can be published after the following revisions:

  1. In the third paragraph, quantitative results of existing methods should be presented to facilitate comparison of research results in this article.
  2. What is the positioning accuracy of the CMM Figure 4 shown?
  3. Image processing is very important for this study, especially for determining the pixel coordinates of the control points, which image processing method was used? What are the parameters? Have sub-pixels been divited?
  4. Line 387, Experiemtn should be Experiment
  5. A space should be left between a number and its unit, such as line 389-391.
  6. Line 335, what is the distance between the two adjacent grids?
  7. Only translations used of the light pen both in calibration and the measurement experiment, how about the results when a rotation occurred in measurement?
  8. Line 388, result->results

Reviewer 2 Report

Calibration is an important and necessary procedure in the Metrology field. Therefore presented paper fits the objectives of the MDPI Sensors Journal.

The paper is divided into five sections.

1. In the Introduction section, the authors presented the needs and challenges for the in-situ geometry measurements. The LPCMS method is discussed together with different calibration techniques.

What would be good to put here in my opinion, is the comparison table of the pros and cons of calibration methods. Could be that provided?

2. Section two describes the LPCMS method and is well presented.

3. Section three is divided into four subsections and presents a complex calibration procedure. It is worth to mention that the authors considered the algorithm optimization by using sparse Jacobian matrices.

4. Section four is an experiment. 

4.1. The characteristic of the light pen is missing.

4.2. How the starting distance of 1.5m between the optical system and the light pe, was measured?

4.3. What type of random errors were added in the simulation part?

4.4. Was the light pen measured by authors before assembly to CMM?

4.5. How room ambient light is affecting measurements?

5. Section five presents the conclusions of the paper. 

The conclusions are well formulated, nevertheless what is missing in my opinion is the summative presentation (e.g. graphical, or tabular) of the accuracy of the presented method in comparison to other methods. The work is partly done in Table 4. If the method is better than other methods, it must be compared to them. Please consider adding it here.

The last comments are just editorial:

Line 34: is "... measurement ..." - should be "... measurement ...".

Line 41: What do you mean by invisible? Do you think of hidden features?

Line 67: is "... the the nonlinear ..." - should be "... the nonlinear ..."

Line 82: is "... translation ..." - should be "... translation ..."

Line 387: is "... Experimnt ..." - should be "... Experiment ..."

Round 2

Reviewer 1 Report

The authors made appropriate revisions to the manuscript based on the comments, the manuscript can be published.